# Choice of Methodology Impacts Outcome in Indirect Comparisons of Drugs for Idiopathic Pulmonary Fibrosis

**DOI:** 10.3390/medicina55080443

**Published:** 2019-08-06

**Authors:** David A. Scott, Emma Loveman, Jill L. Colquitt, Katherine O’Reilly

**Affiliations:** 1Diabetes Research Centre, College of Life Sciences, University of Leicester, Leicester Diabetes Centre, Leicester General Hospital, Gwendolen Road, Leicester LE5 4PW, UK; 2Effective Evidence LLP, 26 The Curve, Waterlooville, Hampshire PO8 9SE, UK; 3Department of Respiratory Medicine, Mater Misericordiae University Hospital, Eccles St., D07 R2WY Dublin, Ireland

**Keywords:** idiopathic pulmonary fibrosis, network meta-analysis, indirect comparisons, nintedanib, pirfenidone

## Abstract

*Background and Objectives:* Idiopathic pulmonary fibrosis (IPF) is a chronic condition leading to lung damage and deterioration in lung function. Following the availability of two new drugs, nintedanib and pirfenidone, a number of network meta-analyses (NMAs) of randomised controlled trials have been published which have conducted indirect comparisons on the two drugs. Differing recommendations from these studies are potentially confusing to clinicians and decision-makers. We aimed to systematically review published NMAs of IPF treatments, to compare their findings and summarise key recommendations. *Materials and Methods:* We systematically reviewed (PROSPERO: CRD42017072876) six eligible NMAs and investigated the differences in their findings with respect to key endpoints. We focused on differences in head-to-head comparisons between nintedanib and pirfenidone. *Results:* The NMAs were broadly consistent, with most differences being explained by model choice, endpoint definitions, inclusion of different studies, different follow-up durations, and access to unpublished data. A substantive difference remained, however, in the change from baseline forced vital capacity (FVC). One NMA favoured nintedanib, another found no statistical difference, whilst others did not conduct the analysis. These differences can be attributed to the choice of methodology, the use of the standardised mean difference (SMD) scale, and population heterogeneity. *Conclusions:* NMA methods facilitated the comparison of nintedanib and pirfenidone in the absence of a head-to-head trial. However, further work is needed to determine whether the trial populations are homogeneous and whether the SMD is appropriate in this population. Differences in patient characteristics may obscure the difference in treatment effects. To assist decision-makers, an exploration of efficacy in real-world populations may be prudent.

## 1. Introduction

Idiopathic pulmonary fibrosis (IPF) is a chronic condition leading to progressive deterioration in lung function and respiratory failure [1]. IPF has a poor prognosis with a median survival from diagnosis of 2–5 years. Two drugs in a new class (anti-fibrotics), pirfenidone and nintedanib, have been shown to slow the deterioration in lung function when compared to placebo. However, there is no direct head-to-head trial evidence of their relative effectiveness to guide prescribers. Hence, comparisons of these treatments have been facilitated through indirect comparisons. 

Published indirect comparisons of pirfenidone and nintedanib have some inconsistencies in their findings and the reasons for these differences have not been examined. We aimed to systematically review published indirect comparisons comparing pirfenidone and nintedanib and explore reasons for the differences in results. The findings elucidate the current uncertainty regarding the relative effectiveness of pirfenidone and nintedanib, and can be used to assist patients, clinicians, and policy-makers with treatment choices.

## 2. Materials and Methods

Accepted methods for searching, study selection, data extraction, and risk of bias assessment were pre-stated in our registered protocol (https://www.crd.york.ac.uk/prospero/display_record.php?RecordID=72876). We identified articles by searches of MEDLINE, EMBASE, and the Cochrane Library and the through checking of reference lists of included studies. There were no language restrictions and searches were conducted by an experienced information scientist. We included indirect comparisons and network meta-analyses (NMAs) of available interventions for people with a confirmed diagnosis of IPF.

Indirect treatment comparisons compare results from trials via a common comparator, maintaining randomisation between treatments within each trial [2]. The principal is illustrated in Figure 1.

In this simple illustration, there are head-to-head trials comparing treatment A to treatment C and treatment B to treatment C, but no trials comparing treatment A to treatment B. Hence, an indirect comparison must be made to compare treatments A and B. 

The relative treatment effect (e.g., log odds ratio or mean difference) for treatment A compared to B is represented by d_AB_*,* treatment B compared to C by d_BC_, and treatment A compared to treatment C by d_AC_. Whilst we have estimates of d_AC_ and d_BC_ from head-to-head trials, we do not know d_AB_. However, under the properties of indirect comparisons this can be calculated by subtracting d_BC_ from d_AC_, as indicated.

NMA extends the indirect comparison concept to networks of trials of direct evidence and indirect evidence, allowing us to add and subtract relative treatment effects to compare all alternative treatments of interest in a single coherent analysis for each outcome [3]. These methods are increasingly being used for healthcare decision-making [4].

NMA can be conducted using frequentist or Bayesian approaches depending upon the software package used. Bayesian analyses use Markov chain Monte Carlo (MCMC) methods, combining prior distributions with the data to construct a posterior distribution upon which to base all summary results [5].

As with traditional pairwise meta-analysis, fixed or random effects models can be used. Whilst variation between studies in a fixed effects model is attributable to statistical chance, in random effect it is attributed to between-study heterogeneity [6]. Heterogeneity can also be adjusted for by including study or population characteristics using meta-regression [7].

The primary outcomes were survival, lung function/capacity, and adverse events. The literature search results were screened by two independent reviewers to identify all citations that may meet the inclusion criteria. Full manuscripts of selected citations were retrieved and assessed by two reviewers against the eligibility criteria. Any disagreements over study inclusion were resolved by consensus.

Two reviewers extracted data from the included NMAs into a previously piloted form to avoid any errors. The methodological quality of the included NMAs was assessed, focusing on the core principles of heterogeneity and inconsistency. The analysis is said to be consistent if the direct and indirect evidence is consistent, i.e., d_AB_ = d_AC_ – d_BC_. Any disagreements between reviewers was resolved by consensus.

The unit of analysis for this systematic overview was the NMA, not the individual trials they contained. Therefore, we did not conduct any new analyses, but conducted a narrative synthesis of the methods, outcomes, and data from the NMAs.

## 3. Results

We searched nine electronic databases and identified 18,125 unique records. After screening titles and abstracts we identified 90 reviews and retrieved full papers for 18 of these that had a pairwise meta-analysis or NMA. Five NMAs published between 2014 and 2017 were included [8,9,10,11,12]. We also identified a poster presentation of a sixth NMA published in 2018 [13]. The randomised controlled trials (RCTs) included by each of the six NMAs are presented in Table 1, and the baseline characteristics of the individual RCTs are summarised in Table 2.

Figure 2 shows the evidence network of 10 RCTs featuring pirfenidone and nintedanib studies. Studies included in the six NMAs differed and two did not connect to the network as there was no common comparator. Commonly reported endpoints were forced vital capacity (FVC), all-cause mortality, and respiratory mortality. The FVC endpoint was measured on two continuous and one binary scale, see Table 1.

Binary endpoints (proportion of patients achieving a >10% decline in FVC, mortality, respiratory mortality, serious adverse events) used the odds ratio (OR) scale, and continuous endpoints (change in % predicted FVC and change in FVC litres) used weighted mean difference (WMD) or standardised mean difference (SMD).

The WMD is a weighted average of the difference in mean treatments effects. The SMD approach is used when trials assess the same outcome but measure it in different ways [23]. In this case the assumption is that the mean change in FVC % predicted is measuring the same thing as litres change in FVC. The SMD approach converts these measures to a common scale. The mean difference between treatment arms is divided by the standard deviation; thus, effect measures are adjusted to be defined in terms of units of SD. SMD thus effectively changes the weights of studies in a meta-analysis. Since difficulties persist in how to interpret treatment effects on the SMD scale, these are converted to ORs using the following formulae from [22]:(1)$$logOR=\frac{\pi }{\sqrt{3}}SMD.$$
(2)$$se(logOR)=\frac{\pi }{\sqrt{3}}se(SMD).$$

Bayesian Markov chain Monte Carlo (MCMC) methods were used in all NMAs.

Statistical heterogeneity in treatment effects was assessed in all NMAs; three discussed similarities across the network of studies and considered them to be sufficiently similar to be combined. One NMA was sponsored by the manufacturer of pirfenidone [9]. 

Choice of methodology differed between the NMAs (Table 3); Fleetwood [9], Rochwerg [12], and Skandamis [13] preferred a random effects model, Loveman et al., 2015 [10] and Loveman et al., 2014 [11] a fixed effects model, whilst Canestaro [8] presented both models.

### 3.1. Forced Vital Capacity (FVC)

Loveman et al., 2015 [10] reported a statistically significant difference in FVC in favour of nintedanib, whilst Fleetwood [9] and Loveman et al., 2014 [11] found no statistically significant difference between treatments. Three NMAs did not report this outcome. There were differences between the NMAs in the length of follow-up included from the trials and whether the FVC was reported as percentage predicted FVC or FVC in litres by the trials (Table 1). The percentage predicted FVC is based on population-based data of individuals of a certain height, age, and gender, and is sometimes additionally corrected for race.

### 3.2. Other Endpoints 

There were no other statistically significant differences between nintedanib and pirfenidone. The proportion of patients with a >10% decline in FVC was consistent across NMAs, with the random effects having a wider credible interval. The all-cause mortality analyses were similar. 

## 4. Discussion

The main sources of differences in the results of the NMAs were the choice of statistical methodology and the data that were included. For FVC, Loveman et al. [10,11] used the SMD approach previously used in IPF by King et al. [24] to combine the change in % predicted FVC and change in FVC (litres). Hence, the underlying assumption is that mean change in FVC % predicted measures the same outcome as litres change in FVC. Fleetwood [9] had access to unpublished mean change in FVC % predicted and change in FVC litres, and thus was able to conducted separate analyses as well as additional follow-up data for all pirfenidone studies from the manufacturer (Table 1). Loveman et al., 2014 [11] was published prior to the availability of more recent data.

For the categorical FVC analyses, differing data were used. Loveman et al., 2015 [10] used data from both Japanese studies of pirfenidone (SP2 and SP3), whereas only one of these (SP2) was used by Canestaro [8] and Fleetwood [9] included neither. Fleetwood used unpublished data for the ASCEND and CAPACITY trials of pirfenidone. Canestaro appeared to pool data from the INPULSIS and CAPACITY studies, which meant those studies were given a higher weight in their fixed effects analysis. Skandamis [13] included two RCTs that were published after searches were undertaken for the other five NMAs [14,15]. These two RCTs do not connect to the network (Figure 2).

The NMAs also included different data for all-cause mortality. For example, Loveman et al., 2015 used conference data from the Capacity trial that was not widely available [10]. Canestaro used data that was not reported in the trial publication of SP2 [8], Fleetwood used reanalysed data of the SP2/SP3 pirfenidone trials studies by Nathan et al. [25], and Canestaro included pooled data from the two INPULSIS studies. Differences are also explained by the use of a random effects model in Fleetwood [9]. Similarly, for respiratory mortality, the higher mean and wider credible intervals in Loveman et al., 2015 [10] compared to Canestaro [8] are explained by the use of use of conference data by Loveman et al., 2015 and the inclusion of pooled INPULSIS studies by Canestaro. Finally, serious adverse events were similar, albeit Skandamis [13] included two additional phase II RCTs.

### Limitations

Sensitivity analysis for the risk of bias of each of the included RCTs was not undertaken, although no studies were considered to have a high risk of bias. 

## 5. Conclusions

The lack of a head-to-head trial comparing nintedanib to pirfenidone has resulted in a number of studies attempting indirect comparison using an NMA approach. For most outcomes, including all-cause mortality, respiratory mortality, and the proportion of patients with a >10% decline in FVC, the published indirect comparisons were broadly consistent in finding no meaningful difference between nintedanib and pirfenidone. Most differences were explained by model choice (fixed or random effects), choice of scale, endpoint definitions, inclusion of different studies, different lengths of follow-up, or access to unpublished data.

A substantive difference between the NMAs remains, however, with respect to conclusions regarding change from baseline FVC. 

Further research is needed to determine:Whether the SMD is appropriate in this population or whether a bivariate approach could be used [26];The functional form of FVC over time to consider the viability of synthesising endpoints across different timepoints;Whether the study populations are sufficiently homogeneous to fit a fixed effect model, whether random effects should be preferred, or whether meta-regression would be plausible;The efficacy of the combined pirfenidone/nintedanib treatment. As this does not connect to the evidence network, a different methodology such as population matching would be required [27].

Finally, differences in patient characteristics may obscure difference in treatment effects, despite the judgement of similarity undertaken by three of the NMAs. A systematic review of prognostic factors in IPF could be conducted to determine the heterogeneity between studies, for example the severity of disease at enrolment. Our review demonstrates how differences in methodological approaches to an NMA can influence outcome. This has potential implications for the use of NMAs in clinical decision-making, in particular with an endpoint such as the FVC. To assist decision-makers, an exploration of efficacy of pirfenidone and nintedanib in real-world populations may be prudent.

## Figures and Tables

**Figure 1 medicina-55-00443-f001:**
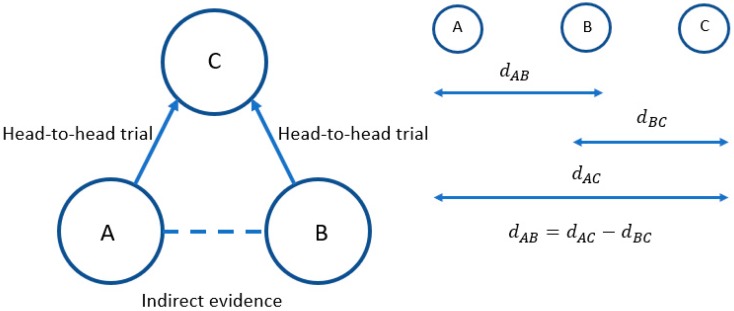
The principles of indirect comparisons.

**Figure 2 medicina-55-00443-f002:**
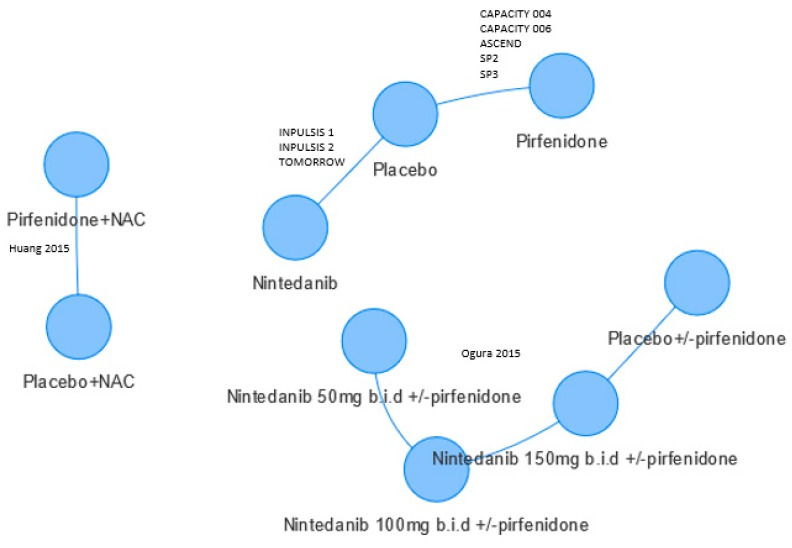
Evidence network. Only Skandamis [13] included Huang 2015 [14] and Ogura 2015 [15]. Some patients in Ogura 2015 received nintedanib and pirfenidone. Circles refer to treatments being compared and may include more than one trial.

**Table 1 medicina-55-00443-t001:** Randomised controlled trials (RCTs) included in each of the network meta-analyses (NMAs) (for at least one outcome).

Relevant RCTs of Nintedanib or Pirfenidone	Inclusion of RCTs (for at Least One Outcome) in the NMAs					
Trial Name, Phase, Forced Vital Capacity (FVC) Outcome, and Timepoints	NMA					
	Fleetwood, 2017 [9] Bayesian Markov chain Monte Carlo (MCMC) Methods, Random Effects	Rochwerg, 2016 [12] Bayesian MCMC Methods, Random Effects	Canestaro, 2016 [8] Bayesian MCMC Methods, Fixed Effects	Loveman 2015 [10] Bayesian MCMC Methods, Fixed Effects	Loveman 2014 [11] Bayesian MCMC Methods, Fixed Effects	Skandamis 2018 [13] (poster only) Bayesian MCMC Methods, Random Effects
SP3 [16], Phase II % predicted at 52 weeks ^a^; Litres at 36 weeks	✓ ^b^	✓	✓	✓	✓	✓
SP2 [17], Phase III % predicted at 52 weeks ^a^; Litres at 52 weeks	✓ ^b^	✓	✓	✓	✓	✓
Capacity 004 [18], Phase III % predicted at 72 weeks; Litres at 48 and 52 weeks ^a^	✓ ^b^	✓^c^	✓ ^c^	✓	✓	✓
Capacity 006 [18], Phase III % predicted at 72 weeks; Litres at 48 and 52 weeks ^a^	✓ ^b^	✓^c^	✓ ^c^	✓	✓	✓
ASCEND [19], Phase III % predicted at 52 weeks ^a^; Litres at 52 weeks	✓ ^b^	✓	✓	✓	Not included	✓
TOMORROW [20], Phase III % predicted at 52 weeks; Litres at 52 weeks	✓	✓	✓	✓	✓	✓
INPULSIS 1 [21], Phase II % predicted at 52 weeks; Litres at 52 weeks	✓	✓^c^	✓ ^c^	✓	Not included	✓
INPULSIS 2 [21], Phase II % predicted at 52 weeks; Litres at 52 weeks	✓	✓^c^	✓ ^c^	✓	Not included	✓
Huang, 2015 [14] Phase II Litres at 48 weeks; % predicted at 48 weeks	Not included	Not included	Not included	Not included	Not included	✓
Ogura, 2015 [15] Phase II Not reported	Not included	Not included	Not included	Not included	Not included	✓

^a^ Unpublished data; ^b^ Utilised unpublished forced vital capacity (FVC) data from the manufacturer; ^c^ Unclear whether Capacity 004 and 006, and INPULSIS 1 and 2 were each included as two separate trials by the NMA.

**Table 2 medicina-55-00443-t002:** Characteristics of RCTs included in the NMAs.

Trial Name, Phase	Intervention, n	Comparator, n	Duration of Treatment	Mean Age	% Male	Time Since Diagnosis	Mean % Predicted FVC	Risk of Bias ^a^
SP3 [16], Phase II	Pirfenidone 1800 mg/day, n = 73	Placebo, n = 36	39 weeks	64	90	<1 year: 22%	80	Unclear
SP2 [17], Phase III	Pirfenidone 1800 mg/day, n = 108	Placebo, n = 104	52 weeks	65	78	<1 year: 37%	78	Unclear
Capacity 004 [18], Phase III	Pirfenidone 2403 mg/day, n = 174	Placebo, n = 174	72 weeks	66	71	≤1 year: 48%	75	Low
Capacity 006 [18], Phase III	Pirfenidone 2403 mg/day, n = 171	Placebo, n = 173	72 weeks	67	72	≤1 year: 59%	74	Low
ASCEND [19], Phase III	Pirfenidone 2403 mg/day, n = 278	Placebo, n = 277	52 weeks	68	78	1.7 years	68	Low
TOMORROW [20], Phase III	Nintedanib 300 mg/day, n = 85	Placebo, n = 85	52 weeks	65	75	1.2 years	80	Low
INPULSIS 1 [21], Phase II	Nintedanib 300 mg/day, n = 309	Placebo, n = 204	52 weeks	67	81	1.7 years	80	Low
INPULSIS 2 [21], Phase II	Nintedanib 300 mg/day, n = 329	Placebo, n = 219	52 weeks	67	78	1.6 years	79	Low
Huang 2015 [14], Phase II	Pirfenidone 1800 mg/day + NAC, n = 38	Placebo + NAC, n = 38	48 weeks	60	93	Not reported	77	Unclear
Ogura 2015 [15], Phase II	Nintedanib ^b^ 100 mg/day, n = 6; 200 mg/day, n = 8; 300 mg/day, n = 24	Placebo ^b^, n = 12	up to 28 days	65	70	Not reported	74	Unclear

^a^ Risk of selection bias. ^b^ A proportion of patients also received pirfenidone. NAC: N-acetylcysteine.

**Table 3 medicina-55-00443-t003:** NMA base case results of nintedanib vs. pirfenidone comparisons (reciprocal calculated where necessary), WMD or OR (95% Crl).

Outcome	NMA					
	Fleetwood, 2017 [9] Bayesian MCMC Methods, Random Effects	Rochwerg, 2016 [12] Bayesian MCMC Methods, Random Effects	Canestaro, 2016 [8] Bayesian MCMC Methods, Fixed Effects	Loveman 2015 [10] Bayesian MCMC Methods, Fixed Effects	Loveman 2014 [11] Bayesian MCMC Methods, Fixed Effects	Skandamis 2018 [13] (poster only) Bayesian MCMC Methods, Random Effects
Change in % predicted FVC	WMD −0.23 (−2.13, 1.66)	Not estimated	Not estimated	**OR 0.67 (0.51, 0.88) ^a^**	OR 0.56 (0.31, 1.03)	Not estimated
Change in FVC Litres	WMD −0.01 (−0.15, 0.13)	Not estimated	Not estimated			Not estimated
>10% decline in FVC	OR 1.11 (0.60, 2.0)	Not estimated	OR 1.16 (0.83, 1.67)	OR 1.21 (0.86, 1.72)	Not estimated	OR 1.10 (0.49, 2.22)
Mortality	OR 1.35 (0.51, 3.70)	OR 1.05 (0.45, 2.78)	OR 1.02 (0.55, 1.89)	OR 1.39 (0.7, 2.82)	Not estimated	OR 1.08 (0.52, 2.63)
Respiratory mortality	Not estimated	Not estimated	1.09 (0.49, 2.38)	OR 2.1 (0.77, 6.17)	Not estimated	Not estimated
Serious adverse events	Not estimated	OR 1.04 (0.51, 2.24)	Not estimated	Not estimated	Not estimated	OR 0.98 (0.62, 1.61)

The statistically significant result is represented by **bold font**; ^a^ converted from standardised mean difference (SMD) to OR using Chinn, 2000 [22]. CrI: credible interval; MCMC: Markov chain Monte Carlo; OR: odds ratio; WMD: weighted mean difference.

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
