# Peer review of "Choice of Methodology Impacts Outcome in Indirect Comparisons of Drugs for Idiopathic Pulmonary Fibrosis"

_medicina, 2019, doi:10.3390/medicina55080443_

Round 1

Reviewer 1 Report

As we know that idiopathic pulmonary fibrosis (IPF) is a chronic condition, which result in lung damage and deterioration in lung function in most of cases. It's a good idea to systematically review published NMAs of IPF treatments, to compare their findings and summarize key recommendations.

it would be great to look at the details in each trials, such the patient population, the stage of disease, gender, age and the treatment options, side effects of medication, allergies and inflammatory reaction; then combined all the data and analyze it by using the same method. Compare the NMAs will be limited by the fixed methods and various of outcome. 

Author Response

We have added a summary table of the included RCTs (Table 2) to assist with the interpretation of whether the fixed or random effects are more appropriate.

Reviewer 2 Report

Since the results on IPF therapy are unclear, the authors wanted to review six network meta analysis  on Pirfenidone and nintenadib, to address the lack of head-to-head trials between the two drugs.

The article is very complicated to understand, and we do not understand the method adopted. The authors also agree that the results are not exhaustive.

Author Response

We have provided further details on our methods. Additional explanatory text and references have been added on indirect comparisons and NMA methodology as well as an illustrative example.

Reviewer 3 Report

Scott and colleagues observed that network meta-analyses (NMA) methodology facilitated comparison of nintedanib and pirfenidone in the absence of a head-to-head trial. However, further work is needed to determine whether the trial populations are homogeneous and whether the standardised mean difference (SMD) is appropriate in this population. In fact, differences in patient characteristics may obscure difference in treatment effects. To assist decision makers, an exploration of efficacy in real world populations may be prudent.

I believe that this is an interesting manuscript because it makes a clear point on the importance, but also on limitations, derived by the utilization of a tool, such as NMA, in comparing the 2 antifibrotic drugs, nintedanib and pirfenidone. As a clinician, the main issue of this study is that it gives us the opportunity to be aware about limitations derived by the use of this methodology. Furthermore, the manuscript is well-written and accurately edited.

Author Response

No action required

Round 2

Reviewer 1 Report

I think that this manuscript is interesting. However, there are some defects, which should be resolved. I don’t have additional comments about the scientific part. I would like authors to recheck the paper for some spelling mistakes and formatting issues.